# StM171, a *Stenotrophomonas maltophilia* Bacteriophage That Affects Sensitivity to Antibiotics in Host Bacteria and Their Biofilm Formation

**DOI:** 10.3390/v15122455

**Published:** 2023-12-18

**Authors:** Ghadeer Jdeed, Vera Morozova, Yuliya Kozlova, Artem Tikunov, Tatyana Ushakova, Alevtina Bardasheva, Andrey Manakhov, Maria Mitina, Elena Zhirakovskaya, Nina Tikunova

**Affiliations:** 1Laboratory of Molecular Microbiology, Institute of Chemical Biology and Fundamental Medicine Siberian Branch of Russian Academy of Sciences, Novosibirsk 630090, Russia; ghadeerjdeed@outlook.com (G.J.); morozova@niboch.nsc.ru (V.M.); ulona@ngs.ru (Y.K.); arttik1986@gmail.com (A.T.); ushakova@niboch.nsc.ru (T.U.); herba12@mail.ru (A.B.);; 2Department of Natural Sciences, Novosibirsk State University, Novosibirsk 630090, Russia; 3Department of Genetics, Centre for Genetics and Life Science, Sirius University of Science and Technology, Sirius 354340, Russia; manakhov@rogaevlab.ru (A.M.); mitina.mi@talantiuspeh.ru (M.M.)

**Keywords:** Caudoviricetes, Stenotrophomonas maltophilia, bacteriophage, comparative genomics, antibiotic resistance, biofilm formation

## Abstract

*Stenotrophomonas maltophilia* mainly causes respiratory infections that are associated with a high mortality rate among immunocompromised patients. *S. maltophilia* exhibits a high level of antibiotic resistance and can form biofilms, which complicates the treatment of patients infected with this bacterium. Phages combined with antibiotics could be a promising treatment option. Currently, ~60 *S. maltophilia* phages are known, and their effects on biofilm formation and antibiotic sensitivity require further examination. Bacteriophage StM171, which was isolated from hospital wastewater, showed a medium host range, low burst size, and low lytic activity. StM171 has a 44kbp dsDNA genome that encodes 59 open-reading frames. A comparative genomic analysis indicated that StM171, along with the *Stenotrophomonas* phage Suso (MZ326866) and *Xanthomonas* phage HXX_Dennis (ON711490), are members of a new putative *Nordvirus* genus. *S. maltophilia* strains that developed resistance to StM171 (bacterial-insensitive mutants) showed a changed sensitivity to antibiotics compared to the originally susceptible strains. Some bacterial-insensitive mutants restored sensitivity to cephalosporin and penicillin-like antibiotics and became resistant to erythromycin. StM171 shows strain- and antibiotic-dependent effects on the biofilm formation of *S. maltophilia* strains.

## 1. Introduction

*Stenotrophomonas maltophilia* is a gram negative, rod shaped, obligate aerobe, and motile bacterium with a few polar flagella [1]. *S. maltophilia* strains display high phenotypic and genetic heterogeneity, forming what is called a *S. maltophilia* complex (SMC) [2]. This bacterium was first isolated in 1943 [3] and is a widespread microorganism found in various environments, including almost all aquatic and humid ones [4,5]. In addition, *S. maltophilia* has been recovered from animals, soils, and plant roots [6,7,8,9]; it also occupies niches inside hospitals, where it demonstrates the ability to form biofilms on medical devices and implants used in patients [1,10,11].

Although *S. maltophilia* is not a highly virulent pathogen, it is associated with a significant mortality rate among immunocompromised patients [5]. *S. maltophilia* can cause different infections, mainly respiratory ones [5]. A number of cases have been recorded among patients with cystic fibrosis and underlying malignancies [12]. *S. maltophilia* has also been isolated from up to 9.4% of blood samples obtained from cancer patients with a bacterial infection [13,14]. As for patients with bacteremia, *S. maltophilia* is associated with mortality rates, ranging from 14% to 69% [1,15,16].

*S. maltophilia* displays intrinsic high-level resistance to a variety of antibiotics and can also acquire resistance through the uptake of the resistance genes located on integrons, transposons, and plasmids via horizontal gene transfer [5,12,17]. Currently, specific guidelines for *S. maltophilia* treatment by antibiotics are absent [5,13]. The choice of antibiotics to treat *S. maltophilia* infections is often based on conventional antimicrobial testing using a plankton bacterial culture, whereas *S. maltophilia* is clinically relevant due to its ability to form biofilms. Biofilms formed by *S. maltophilia* are known to be the starting point of various chronic infections and exhibit greater resistance to antibiotics than non-biofilm forming bacteria and, therefore, more difficult to treat. *S. maltophilia* can form biofilms both on abiotic surfaces and host tissues, dramatically enhancing the resistance to therapeutically important antibiotics, including aminoglycosides, fluoroquinolones, and tetracycline [18,19]. It is therefore suggested that the selection of antibiotics for S. maltophilia infections based on biofilm sensitivity testing could provide an accurate and effective guide on appropriate treatments [20]. *S. maltophilia*’s intrinsic resistance to different antibiotics and its ability to develop resistance to new ones and form biofilms led the World Health Organization to list it as one of the leading drug-resistant pathogens in hospitals worldwide [10,21].

Bacteriophages (phages) represent a promising option for treating resistant bacteria, either in combination with antibiotics or on their own, and in most cases, a cocktail composed of several phages is used. To date (September 2023), 60 *S. maltophilia* phages were isolated, with some of them showing a broad range of lytic efficiency [22], while others were lysogenic [23]. In this study, we present biological properties of *S. maltophilia* phage StM17, including the host range and its genome analysis. We also describe changes in antibiotic sensitivity of bacterial StM17-insensitive mutants and the effect of this phage on biofilm formation of host cells in combination with different antibiotics. In addition, we analyze the genomes of five *S. maltophilia* strains susceptible to StM171 and focus on the genotype related to antibiotic resistance and biofilm formation.

## 2. Materials and Methods

### 2.1. Strains Source and Culture Conditions

Bacterial strains used in this study were obtained from the Collection of Extremophilic Microorganisms and Type Cultures (CEMTC) of the Institute of Chemical Biology and Fundamental Medicine, Siberian Branch of the Russian Academy of Sciences (ICBFM SB RAS) (Appendix A). The strain *S. maltophilia* CEMTC 2355, which was used as the host for initial phage isolation, plaque assay, and propagation of phage, was isolated from hospital wastewater in Novosibirsk, Russia. Strains that were applied for biofilm and antibiotic studies included *S. maltophilia* CEMTC 2142 (isolated from wastewater) and the strains *S. maltophilia* CEMTC 3659, CEMTC 3664, and CEMTC 3670, which were isolated from insects in Novosibirsk, Russia. The strains were identified as *S. maltophilia* by sequencing a 1308 bp fragment of the 16S rRNA gene, as described previously [24], and 16S rRNA sequences of the investigated strains were deposited in the GenBank database under accession numbers MZ424754, OP393915, MT040043, MT040044, and MT040045 for the strains *S. maltophilia* CEMTC 2142, CEMTC 2355, CEMTC 3659, CEMTC 3664, and CEMTC 3670, respectively. *S. maltophilia* strains were cultivated in Luria–Bertani broth (Thermo Fisher Scientific, Waltham, MA, USA) or using Beef meat enzymatic agar (Pharmacotherapy Research Center, Saint Petersburg, Russia) and fish peptone agar (Microgen, Moscow, Russia) aerobically at 37 °C. Bacterial growth was monitored by measuring optical density at 600 nm (OD600). For *S. maltophilia* 2355, an OD600 of 1.0 corresponded to 8 × 10^8^ cells/mL.

### 2.2. Phage Isolation and Purification

StM171 phage was isolated from hospital wastewater using host strain *S. maltophilia* CEMTC 2355. Bacteriophage isolation and propagation were performed as described previously [25]. Briefly, the sample was centrifuged (5000× *g*, 10 min at 4 °C), and the supernatant was filtered using a PES Welded 0.22 mm syringe-driven filter (Labfil, Hangzhou, China) and stored at 4 °C. The presence of phages was checked by spotting 10 µL aliquots of the supernatant on the double-layered plates containing the lawns of several *S. maltophilia* strains. A part of the top agar containing a separate plaque was picked and mixed with PBS before repeating centrifugation and filtration. After appropriate dilution, the obtained supernatant aliquots were plated for plaque formation. At least two more successive single-plaque isolations were performed to obtain the phage isolate.

A single plaque was picked to propagate a working-stock solution for analysis using top-agar overlays. Briefly, 300 µL of the overnight *S. maltophilia* CEMTC 2355 culture containing ~10^9^ plaque-forming units per ml (PFU/mL) were added to 3 mL of 0.7% LB top agar. The mixture was poured onto the LB plate and incubated for 18 h at 37 °C. The top agar of plates showing confluent lysis was scraped into a 50 mL Falcon tube. A 3 mL aliquot of STM-buffer (10 mM NaCl, 50 mM Tris-HCl, pH 8.0, 10 mM MgCl_2_) was added for each plate scraped, and the suspension was shaken for 1 min followed by centrifugation (5 min at 10,000× *g*) and 0.22 mm filter sterilization.

### 2.3. Transmission Electron Microscopy

A carbon-coated copper grid was overlaid with a drop of phage suspension for 1 min and then stained with 1% uranyl acetate for 5–7 s. A JEM 1400 transmission electron microscope (JEOL, Tokyo, Japan) was used to obtain electron microscopy images of StM171. Digital images were collected using a side-mounted Veleta digital camera (Olympus SIS, Hamburg, Germany). The capsid diameter, tail length, and tail width of virions were measured using ImageJ (version 1.51) [26], and average sizes were calculated using Microsoft excel 365.

### 2.4. One-Step Growth Curve

One-step growth experiments were performed as described previously [27] with some modifications. *S. maltophilia* CEMTC 2355 cells were cultivated (OD600 = 0.4), harvested by centrifugation, and re-suspended in fresh LB (4.0 × 10^9^ colony-forming units per mL, CFU/mL). Phage StM171 was added at a multiplicity of infection (MOI) of 0.001 and allowed to adsorb for 5 min at 37 °C. The mixture was centrifuged (6000 rpm, 7 min), and the pellets containing infected cells were suspended in 10 mL of LB, followed by incubation at 37 °C. Samples were taken at 15 min intervals (up to 3.5 h), diluted, and titers were determined by the double-layered agar plate method.

### 2.5. Kinetics of Cell Culture Lysis

The kinetics of cell culture lysis experiment were performed as follows: *S. maltophilia* CEMTC 2355 cells (OD600 = 0.4) were infected with the phage StM171 at MOI of 0.01 and allowed to adsorb at 37 °C for 30 min with no shaking followed by incubation with shaking. Aliquots (100 µL) were taken at 30 min interval (up to 6 h). Each aliquot was diluted, 10 µL were applied on plates with LB agar, and individual bacterial colonies were counted next day.

### 2.6. Phage Host Range Analysis

Host range analysis was performed using serially diluted StM171 lysate (concentrations of 10^9^–10^5^ PFU/mL). A five µL aliquot of each dilution was spotted in triplicate onto a plate containing bacterial culture in a top-agar overlay and incubated overnight at 37 °C. Ten *S. maltophilia* and twenty-six *P. aeruginosa* bacterial strains from the CEMTC of the ICBFM SB RAS were used for host range examination (Appendix A).

### 2.7. StM171’s DNA Isolation and Sequencing

Genomic DNA was isolated from a StM171 stock (10^7^ PFU/mL) as described previously [28]. Phage particles were precipitated using a PEG/NaCl solution and dissolved in an STM-buffer (10 mM NaCl, 50 mM tris-HCl, pH 8.0, 10 mM MgCl_2_). The obtained phage suspension was supplemented with 15 units DNase I (Thermo Fisher Scientific, Waltham, MA, USA) in DNase I buffer (10 mM tris-HCl pH 7.5, 2.5 mM MgCl_2_, 0.1 mM CaCl_2_), up to 20 µg/mL RNase (Thermo Scientific, Waltham, MA, USA), and incubated for 30 min at 37 °C. Then, SDS (final concentration of 0.5%), up to 20 mM of 0.5 M EDTA, pH 8.0, and Proteinase K (up to 200 µg/mL) were added, followed by incubation at 55 °C for 3 h. Phage DNA was purified using phenol chloroform extraction with following ethanol precipitation. A paired-end library of phage StM171 DNA was prepared using the NEBNext Ultra II DNA Library Prep Kit for Illumina (New England Biolabs, Inc., Ipswich, MA, USA). Sequencing was carried out using the MiSeq Benchtop Sequencer and MiSeq Reagent Kit v.2 (2 × 250 base reads). The genome was assembled de novo by SPAdes v.3.15.4 and resulted in one genomic contig with an average coverage of 150×.

### 2.8. Phage Genome Analysis

Putative open reading frames (ORFs) were determined and annotated using RAST [29,30,31] and checked manually against NCBI GenBank database using Blastx (version 2.12.0) [32]. I-Tasser server (version 2.1) [33] and HHpred (version 3.2.0) [34] were used to determine function for hypothetical proteins that were not found in NCBI GenBank. Phage genome was searched for tRNA using tRNAscan-SE software (version 2.0) [35]. Proksee genome analysis server [36] was used for comparative analysis (using BLAST+ 2.12.0 tool [37]) of the StM171 genome against the genomes of the closest phages (*Stenotrophomonas* phage Suso, MZ326866, and *Xanthomonas* phage HXX_Dennis, ON711490) and visualization was performed using CGView builder (version 1.1.2) [38]. VectorBuilder’s GC Content Calculator [39] was used to determine the GC content of the phage.

### 2.9. Phylogenetic Analysis

The genetic distance between genomes was calculated using MEGA 11 software [40]. Comparative proteomic analysis was performed using ViPTree [41] in comparison with the RefSeq version 216 database with addition of 50 *S. maltophilia* phages from the NCBI GenBank database. Blastx [32] was used to find the closest proteins to the characteristic proteins of StM171 phage (capsid, tail fiber, tape measure, large terminase subunit) in the NCBI GenBank database (accessed August 2023). Mega 11 was used to perform multiple sequence alignment following MUSCLE method and to build phylogenetic trees using Maximum Likelihood estimation with bootstrap 1000. vConTACT2 [42] was used to compare StM171 with 3552 genomes of prokaryotic viruses from the NCBI Refseq version 216 database with International Committee on Taxonomy of Viruses (ICTV) and NCBI taxonomies in addition to all found *S. maltophilia* phages (April 2023). Based on the number of shared protein clusters (PCs) between the genomes, vConTACT2 calculated the degree of similarity as the negative logarithmic score by multiplying the hypergeometric similarity *p* value by the total number of pairwise comparisons. Subsequently, Markov cluster (MCL) was chosen for protein clustering and pairs of closely related genomes with a similarity score of ≥1 was grouped into viral clusters (VCs) created by clustering with overlapping neighborhood expansion (ClusterONE). A protein sharing network was visualized using Cytoscape (version 3.9.0) [43]; the model chosen for the visualization was an edge-weighted spring embedded model, which placed the genomes or fragments sharing more PCs closer to one another. In the visualization, each node represented a phage, whereas the length of the edge between nodes was based on the similarity of the gene content between each pair of genomes. Only *Stenotrophomonas* phage nodes and nodes directly linked to them were displayed.

### 2.10. Bacterial Host Genome Sequencing and Analysis

*S. maltophilia* genome DNA was purified using DNA isolation kit (Biolabmix, Novosibirsk, Russia). The obtained DNA was diluted in 10% TE-buffer and its concentration was measured by Qubit 4.0 (Thermo Fisher Scientific, Waltham, MA, USA). Purified DNA was fragmented with ultrasonic disintegrator Covaris S220 (Covaris, Woburn, MA, USA) to the average size 500 bp. Genomic libraries were synthesized using TruSeq DNA PCR-Free kit and TruSeq Illumina adaptors (New England Biolabs, Inc., Ipswich, MA, USA). KAPA Library Quantification Kit (KAPA Biosystems, Wilmington, MA, USA) was used to quantify all synthesized libraries in order to mix them in equimolar concentrations in one pull. This pull of libraries was sequenced on Illumina NovaSeq 6000 (Illumina, San Diego, CA, USA) with NovaSeq 6000 S2 Reagent Kit (version 1.5) (2 × 150 cycles). Raw data were demultiplexed with bcl2fastq v2.20.0.422 using default parameters. Filtration and adapter removal were performed with Trimmomatic (version 0.39) These trimmed reads were used to assemble genome contigs using SPAdes (version 3.15.4)

The genomes of *S. maltophilia* bacterial strains were analyzed using Bactopia (version 2.2.0) [44] workflow, the following modules and analysis were utilized and conducted: AMRFinderPlus (version 3.10.45) [45,46] and ABRicate (version 1.0.0) [47] to screen for antimicrobial resistance genes in the following databases: ARG-ANNOT (version 6) [48], CARD (version 3.2.8) [49], ResFinder (version 4.1.5) [50], VFDB (accessed 2023.04.17) [51], MEGARes (version 2.0) [52], and Bacterial Antimicrobial Resistance Reference Gene Database (accessed 2023.04.17) [45]. Technical quality of the sequences was assessed using QUAST (version 5.2.0) [53]; genome distance estimation was performed using Mash (version 2.3) [54] and sourmash (version 4.7.0) [55]; for quality control, BBTools (version 39.03) [56] and Lighter (version 1.1.2) [57] were used to clean up the FASTQs; FastQC (version 0.12.1) [58] and Fastq-Scan (version 1.0.1) [59] were used for summary statistics before and after cleanup; and Prokka (version 1.14.5) [60] and RAST were used for annotation, server. The comparative results were visualized using Circos (version 0.69.8) [61] hosted on The Galaxy platform (version 23.2.rc1) [62].

### 2.11. Antibiotic Resistance of S. maltophilia Strains

The sensitivity of bacterial strains (*S. maltophilia* CEMTC 2142, 2355, 3659, 3664, and 3670) to 23 different antimicrobials was assayed using a disk diffusion assay (OXOID, Basingstoke, UK). The diameter of the bacterial growth inhibition zone around the disk was a criterion of resistance. The Minimum Inhibitory Concentration (MIC) for some of the antibiotics was determined by applying decreasing concentrations of antibiotics to planktonic *S. maltophilia* cells. The full list of antimicrobials tested and their MIC are in Appendix A, respectively.

### 2.12. Anti-Biofilm Activity of StM171 with and without Antibiotics

Biofilm formation ability of five *S. maltophilia* strains susceptible to StM171 (*S. maltophilia* CEMTC 2142, 2355, 3659, 3664, and 3670) was determined as described previously [63] with some modifications. Briefly, bacterial strains were incubated on Mueller Hinton Agar (Oxoid, UK) overnight at 37 °C, and the obtained colonies were suspended in Luria–Bertani broth (Becton, Dickinson and Company, Sparks, MD, USA) to OD600 = 0.5. Then, bacterial suspensions were incubated in TSB (glucose 1%) liquid media in 96-well plate for 24 h at 37 °C. The wells were washed with PBS, dried at 60 °C, and stained with Crystal Violet (Carl Roth, Karlsruhe, Germany). The optical density was measured with a spectrophotometer (iMark, Biorad, Tokyo, Japan); the experiment was performed in triplicates.

To test the effect of StM171 on biofilm formation, the same protocol was used with the addition of StM171 at a final MOI = 0.1 and dilution of the bacterial suspensions in LB to OD600 = 0.2. The experiment was performed in five replicates. The experiments in combination of different antibiotics with StM171 were performed by applying the same protocol with the addition of the following antibiotics: ampicillin (toward which the strains are resistant), chloramphenicol, levofloxacin, tetracycline, and gentamicin at a final concentration of 40% of MIC for them against the appropriate *S. maltophilia* planktonic culture. The experiment was repeated five times.

### 2.13. Propagating Bacterial Strains with Resistance to StM171 and Retesting Their Resistance to Antibiotics

Single colonies of bacterial strains that developed resistance toward StM171 phage after applying it in double layer assay were picked, propagated, and tested three times in a row to ensure that StM171 lost the ability to infect them. To check for the presence of the StM171 genome in *S. maltophilia* bacterial strains, two sets of primers that targeted the genes encoding the capsid and tail assembly protein were designed. Primers are in Appendix A. Bacterial strains were heated at 95 °C for 10 min before centrifugation at 14,000 rpm for 5 min, and samples were taken from the supernatant to be used as a template for PCR. To check the effects of the antibiotics on the bacterial clones, a total of 13 antibiotics (gentamicin, ampicillin, amoxicillin, ampicillin with sulbactam, cefoxitin, ceftazidime, cefepime, chloramphenicol, levofloxacin, ciprofloxacin, clindamycin, tetracycline, and trimethoprim with sulfamethoxazole) were tested on those strains, and the original strains using disk diffusion assay. The experiment was performed in triplicates in three repeats.

### 2.14. Statistics

Statistical analysis was performed using paired *t*-test in RStudio, cut-off *p* value (0.05), and in Microsoft Excel (version 2110).

## 3. Results

### 3.1. Characteristics of StM171 and Its Growth Dynamics

Transmission electron microscopy showed that StM171 has an elongated capsid (47.3 nm ± 0.1 nm × 45.7 nm ± 2 nm) and a flexible non contractile tail (~170 nm) indicating that phage StM171 belongs to the order Caudoviricetes and has a siphovirus morphology (Figure 1 and Appendix A).

Kinetics of cell culture lysis experiment indicated that StM171 has a weak lytic activity, causing a maximum of 1-fold reduction in the log of CFU after prolonged incubation (Figure 2A). However, the plaques formed by StM171 in a double layer agar were clear, with a diameter of 0.6 mm ± 0.1 mm. The average burst size of StM171 was calculated as the ratio of the final count of released phage particles to the initial count of infected bacterial cells, and it was approximately 12 PFU/cell (Figure 2B). The latent period was ~130 min when infecting the host strain *S. maltophilia* CEMTC 2355 grown in LB medium at 37 °C.

StM171 was tested against 10 *S. maltophilia* strains and 74 *Pseudomonas aeruginosa* strains from CEMTC of the ICBFM SB RAS using a double layer agar method. The StM171 phage showed a moderate host range against *S. maltophilia*, forming clear plaques on five strains (*S. maltophilia* CEMTC 2142, CEMTC 2355, CEMTC 3659, CEMTC 3664, and CEMTC 3670), and it did not infect any of the tested *P. aeruginosa* strains (Appendix A)

### 3.2. StM171 Genomics and Genome Organization

StM171 genome is a dsDNA composed of 44,512 bp; it contains 59 predicted ORFs. Among the ORF, 35 were assigned putative functions, whereas other ORFs encoded hypothetical proteins that showed no homology to other characterized sequences (Figure 3). The StM171 genome has a GC content of 67.32%, which is close to those of susceptible *S. maltophilia* strains (66.45%, 66.37%, 66.52%, 66.55%, and 66.3% for *S. maltophilia* strains CEMTC 2142, CEMTC 2355, CEMTC 3659, CEMTC 3664, and CEMTC 3670, respectively).

StM171 genome includes the genes encoding proteins responsible for DNA metabolism, DNA packaging, cell lysis, and structural proteins. Genes that encode DNA- or RNA-polymerases were not found in the StM171 genome, apparently, the phage uses the bacterial host polymerases. No known genes (e.g., integrase or transposase) for the temperate lifestyle were detected despite StM171 has a siphovirus morphology. The StM171 phage genome was deposited in the NCBI GenBank database under accession number MZ611865.

A comparative analysis of the StM171 genome with other phage genomes indicated the highest similarity with two phages, namely *Stenotrophomonas* phage Suso and *Xanthomonas* phage HXX_Dennis (GenBank accession numbers MZ326866 and ON711490, respectively). StM171 shares high level of nucleotide identity with phages Suso and HXX_Dennis (95.03% and 95.47%, respectively) making them members of one species. Notably, these three phages were isolated from three different countries: the USA, Canada, and Russia. Additionally, a metagenome-assembled genome (GenBank accession number SSEB01000020) from California, USA, showed a >95% identity to the phage StM171 (Figure 3).

### 3.3. StM171 Phylogenetic Analysis

According to ViPTree analysis, StM171 is located on a distinct branch with phages Suso and HXX_Dennis (Figure 4).

This branch is part of a clade formed by a large group of Mycobacterium phages belonging to the *Bclasvirinae* subfamily and a few *Gordonia* and *Rhodococcus* phages from *Phrappuccinovirus*, *Skogvirus*, and *Puppervirus* genera.

A maximum likelihood phylogenetic analysis of several StM171 proteins (terminase large subunit, tail tape measure protein, capsid protein, and tail fiber protein) and the closest analogs from other phages indicated that the phages with the highest homology to StM171 were different than those suggested by ViPTree (Figure 5). However, StM171 proteins still form highly supported branches with corresponding sequences from the *Stenotrophomonas* phage Suso and *Xanthomonas* phage HXX_Dennis (Figure 5).

In addition, VConTACT2 [42] was used to create a viral cluster network where the phages are positioned based on the similarity of the genes, and visualization of the created networks with the Cytoscape program (version 3.9.0) [43] showed that StM171 did not belong to any known genus, being grouped with the phages Suso and HXX_Dennis (Figure 6). The phages that showed the highest similarity belonged to the following genera: *Beetrevirus*, *Casadabanvirus*, *Rosemountvirus*, *Yuavirus*, *Pamexviurs*, *Nipunavirus*, *Seurtavirus*, *Septimatrevirus*, and *Amoyvirus* that contain mainly *Pseudomonas*, *Stenotrophomonas*, and, to a lesser degree, *Salmonella*, *Achromobacter*, and *Escherichia* phages (Figure 6). A complete table of calculated similarity scores of StM171 and other phages is provided in Appendix A.

According to the analysis, we assume that StM171 belongs to a new genus within the Caudoviricetes order, along with the *Stenotrophomonas* phage Suso and the *Xanthomonas* phage HXX_Dennis. The proposed name for the genus is the *Nordvirus*.

### 3.4. Analysis of the StM171 Bacterial Host Genomes

The genomes of five bacterial hosts of StM171 were sequenced and analyzed. They were classified based on their isolation site into two groups: group A strains (*S. maltophilia* CEMTC 2142 and CEMTC 2355) were isolated from wastewater, and group B strains (CEMTC 3659, CEMTC 3664, and CEMTC 3670) were isolated from insects. Group B strains displayed a higher level of genotype and phenotype similarity compared to the group A strains.

All the strains possessed the genes *smf-1*, *rpfF*, *rpfG*, and *rpfC*, which are involved in biofilm formation of *S. maltophilia* [64,65]. The proteins encoded by the group B genes were identical, whereas corresponding proteins from the group A strains differed from them by a number of amino acid residues (aa), RpfC, RpfF, and Smf-1; proteins of *S. maltophilia* CEMTC 2142 differed by 3, 1, and 5 aa, respectively; and those of` *S. maltophilia* CEMTC 2355 differed by 6, 2, and 2 aa, respectively.

Eighteen antibiotic resistance genes were found in the five studied strains, the exception was the β-lactams resistance gene *aac(6′)-Iz* [66] that was found only in the group A strains. Notably, the antibiotic resistance genes were identical in group B strains, while twelve out of eighteen antibiotic resistance genes in group A showed variations in their nucleotide sequences (Figure 7).

The antibiotic resistance genes that were found included the following: *emrRsm*, *emrCsm*, *emrAsm*, and *emrBsm* genes, which form a four-member operon regulated by EmrRsm. The *emrABCRsm* operon is a multidrug resistance pump that protects the bacteria from several chemically unrelated antimicrobial agents, including fluoroquinolone antibiotics [67]. In addition, the *smeA*, *smeB*, and *smeC* genes that encode the efflux pump SmeABC as well as *smeS* and *smeR* genes that encode two component transduction system smeSR were revealed. The SmeABC efflux pump and SmeSR transduction system are linked to resistance to aminoglycosides, beta-lactams, and fluoroquinolones [68]. Other antibacterial resistance genes found were *smeD*, *smeE*, and *smeF* that are responsible for the efflux pump SmeDEF regulated by the repressor gene *smeT*. SmeDEF determines resistance to quinolones, tetracyclines, macrolides, chloramphenicol, and novobiocin [69]. Finally, the following antibacterial resistance genes were revealed: the beta lactamase genes *bla1* and *bla2*, which are linked to resistance to penicillin, cephalosporin, and carbapenem [70,71,72]; *aph(3’)-Iic* and *aph(6)-Smalt* genes that are associated with resistance toward aminoglycosides [73]; and the gene *oqxB9*, which encodes the RND efflux pump responsible for resistance to fluoroquinolone [74] (Figure 7). A detailed list of the antibiotic resistance genes found in StM171 bacterial hosts is provided in Appendix A.

### 3.5. Changes in Antibiotic Sensitivity in S. maltophilia Clones That Developed Resistance to StM171

The use of StM171 against host *S. maltophilia* strains led to the appearance of bacterial StM171-insensitive mutant (BIM) clones. Phage resistance in these clones persisted after at least three passages. This effect was observed for all five host StM171 strains. PCR was used to confirm the lack of integration of StM171 DNA into the genomes of five BIMs corresponding to host strains.

Thirteen antibiotics were tested against the StM171 resistant clones and compared with their phage-sensitive ancestors. Currently, there are no EUCAST recommendations regarding the evaluation of *S. maltophilia* sensitivity to antibiotics using the disk diffusion method (with the exception of trimethoprim/sulfamethoxazole). As an indicator of antibiotic resistance, we used the diameter of the clean/clearing zone around the disk, free of bacteria. When this value was 5–6 mm, it was considered that the tested strain was resistant to this antibiotic.

The obtained results indicated that StM171-resistant strains from Group B became sensitive to penicillin-like antibiotics (ampicillin, ampicillin-sulbactam, amoxicillin) and cephalosporins (cefepime and ceftazidime) (Figure 8, Appendix A). On the contrary, these strains gained resistance to erythromycin, whereas StM171-resistant strain *S. maltophilia* CEMTC 2142 acquired resistance to cefepime. Resistance of the studied *S. maltophilia* strains to trimethoprim/sulfamethoxazole remained independent of resistance to StM171; resistance to other tested antibiotics did not change significantly (Figure 8, Appendix A).

### 3.6. Effect of StM171 with and without Antibiotics against Biofilm Formation

The ability of the five StM171-susceptible *S. maltophilia* strains to form biofilms was tested, and it was shown that they were able to form biofilms. The ability of StM171 to inhibit biofilm formation was investigated, and this phage demonstrated a strain-dependent effect (Figure 9). StM171 reduced biofilm formation in *S. maltophilia* CEMTC 2142, increased the formation of biofilm in the case of *S. maltophilia* CEMTC 3670, and had no significant effect on the other three strains (*S. maltophilia* CEMTC 2355, *S. maltophilia* CEMTC 3659, and *S. maltophilia* CEMTC 3664).

When antibiotics were used at 40% of their minimum inhibitory concentrations (MIC) against a bacterial host without StM171; chloramphenicol (except, again, *S. maltophilia* CEMTC 2142), levofloxacin, tetracycline, and gentamicin significantly inhibited the formation of biofilm by all the strains, whereas ampicillin expectedly did not affect biofilm formation (Figure 9). If tested antibiotics were combined with StM171, the effect was divergent. StM171 enhanced the inhibitory effect of antibiotics in most cases; however, the only effect of tetracycline and levofloxacin on biofilm formation by *S. maltophilia* CEMTC 2355 and *S. maltophilia* CEMTC 3664, respectively, was significant. Notably, the effect of gentamicin on biofilm formation in the case of group B strains was somewhat weakened by StM171 (Figure 9).

## 4. Discussion

In this study, *S. maltophilia* phage StM171 was isolated and characterized. It represents a novel genus *Nordvirus* within the Caudoviricetes class together with the *Stenotrophomonas* phage Suso and *Xanthomonas* phage HXX_Dennis. Despite being isolated in different continents, all these phages belong to the same species due to the nucleotide identity of >95%. Although StM171 has siphoviral morphology, no known genes responsible for the temperate lifestyle of StM171 were found.

The unusual divergence in the closest genera between ViPTree and vConTACT2 analyses may be the result of different bioinformatic algorithms used. Notably, the phylogeny of signature proteins (terminase large subunit, major capsid protein, tail fiber protein, and tape measure protein) also suggests divergence of neighboring genera. However, members of the proposed *Nordvirus* genus are grouped together in an all-comparative analysis. The obtained results indicated that members of this genus display a high level of mosaicism and that the evolution of this genus was complicated.

Five *S. maltophilia* strains sensitive to StM171 were identified. These hosts were divided into two groups depending on their genetics and place of isolation. Group A included two strains isolated from wastewater, whereas group B contained three strains isolated from insects. Group B strains were more similar on the genotype and phenotype levels, especially in terms of antibiotic resistance and the restoration of antibiotic sensitivity. We assume that the gained resistance to erythromycin in BIMs (strains with resistance to StM171) that descended from the group B strains was due to changes in the cell surface that led to resistance to StM171 and prevented the penetration of erythromycin [75,76]. As for the restoration of sensitivity to cefepime and ampicillin in group B strains, we should note that the *bla2* genes that are linked to resistance to these antibiotics [70,71,72] were identical in these strains and differed from group A strains. This fact may indicate a possible role for the regulation of these genes; however, further investigations are required.

In conclusion, we identified the *S. maltophilia* phage StM171 with low lytic activity; however, even a low-active phage can increase or even restore the sensitivity of host bacteria to certain antibiotics. When host sub-strains with temporal resistance to StM171 appear, they may have both decreased and increased sensitivity to antibiotics, which depends on the antibiotic and the bacterial strain. The study of the effect of phages on antibiotic resistance and biofilm formation is an urgent problem, which makes it important to further study this interaction when applying phages against specific bacteria.

## Figures and Tables

**Figure 1 viruses-15-02455-f001:**
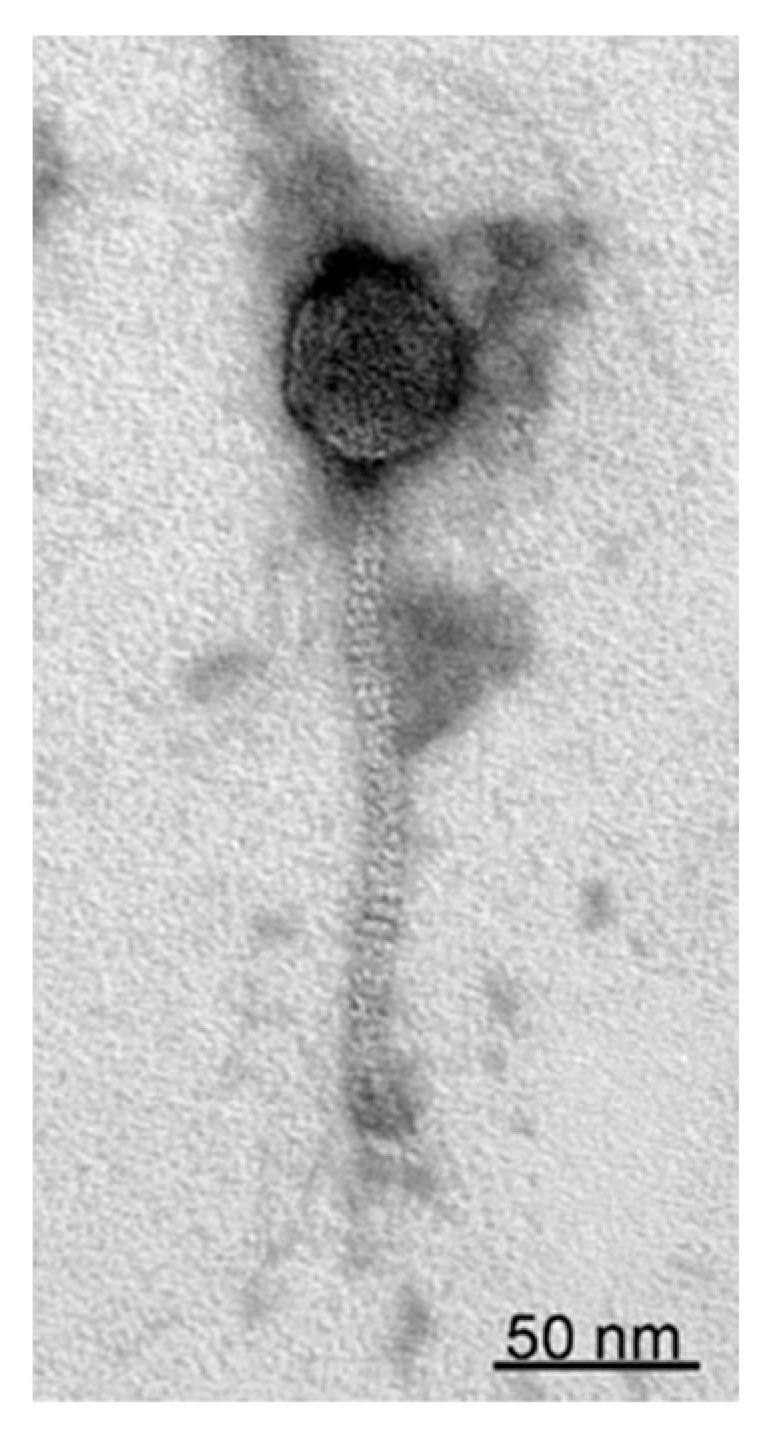
Transmission electron microscopy of the StM171 phage particle negatively stained with 1% uranyl acetate.

**Figure 2 viruses-15-02455-f002:**
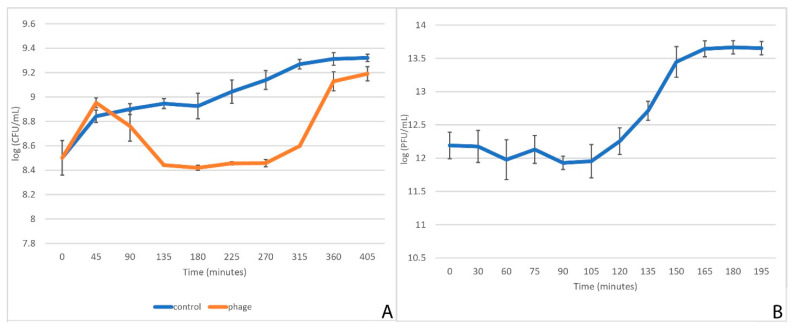
StM171 biological characteristics upon *S. maltophilia* CEMTC 2355. (**A**): kinetics of lysis, in which bacterial cultures incubated with phage StM171 (phage) were compared with non-infected bacterial cultures (control). (**B**): one step growth curve of StM171 on host strain *S. maltophilia* CEMTC 2355.

**Figure 3 viruses-15-02455-f003:**
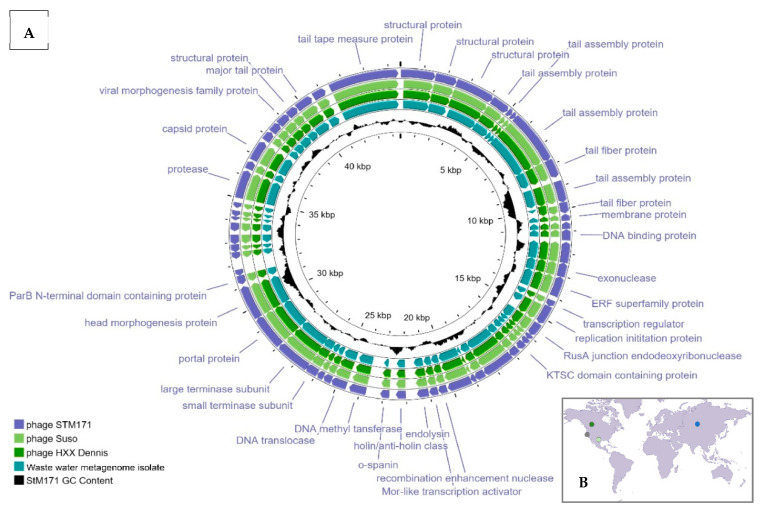
(**A**) StM171 phage genome features, compared with those of the phages with the highest nucleotide identity *Stenotrophomonas* phage Suso (MZ326866), *Xanthomonas* phage HXX_Dennis (ON711490), and a wastewater metagenome isolate (SSEB01000020.1); the figure was constructed using Proksee server. (**B**) A map produced using Canva depicting the isolation sites of the above phages: Novosibirsk, Russia (StM171); Austin, USA (Suso); Edmonton, Canada (HXX_Dennis); California, USA (wastewater metagenome isolate).

**Figure 4 viruses-15-02455-f004:**
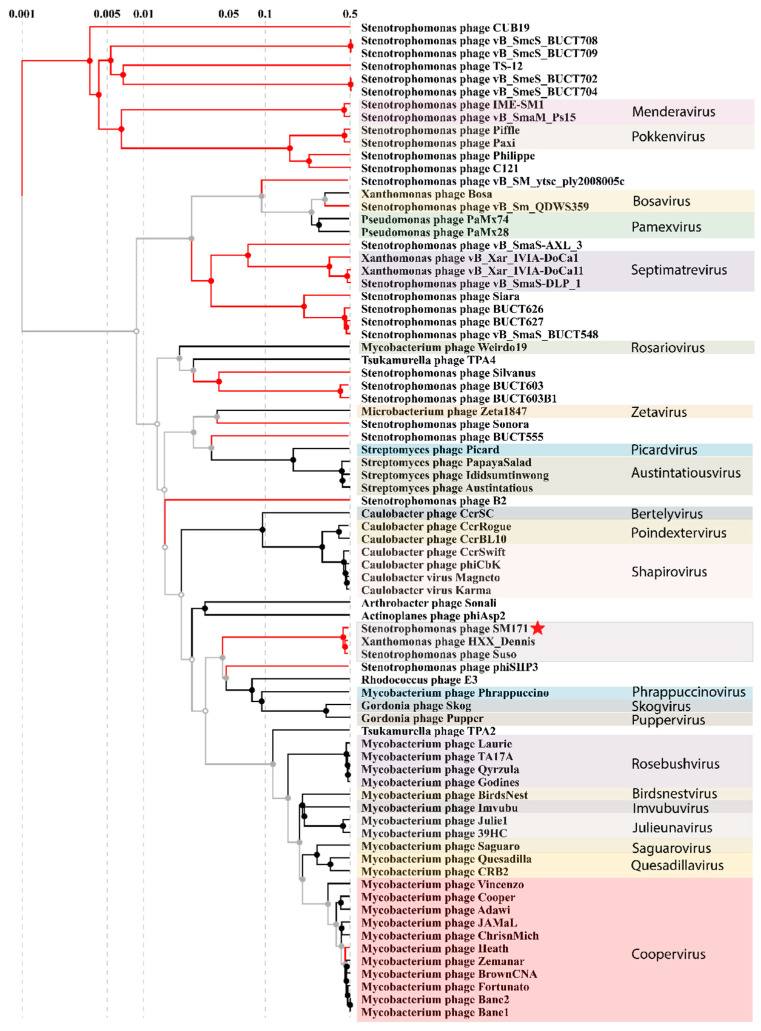
Proteomic tree created with ViPTree server indicates the phages closest to StM171 and their genera. Red branches indicate phage sequences that were downloaded from the NCBI GenBank database and added to the analysis manually. The studied phage StM171 is marked with a red asterisk.

**Figure 5 viruses-15-02455-f005:**
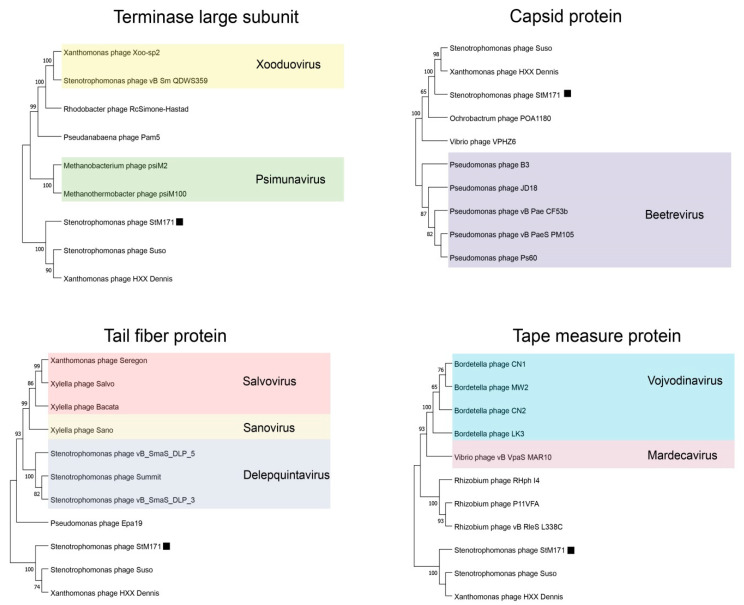
Phylogenetic trees created using Maximum Likelihood estimation with bootstrap 1000 of the aligned sequences of StM171 proteins and close analogs from other phages. The close genes were found using BLASTX, the sequences were aligned using Muscle method, the alignments and phylogenetic tree analysis were performed in MEGA 11. The studied phage StM171 is marked with a black square.

**Figure 6 viruses-15-02455-f006:**
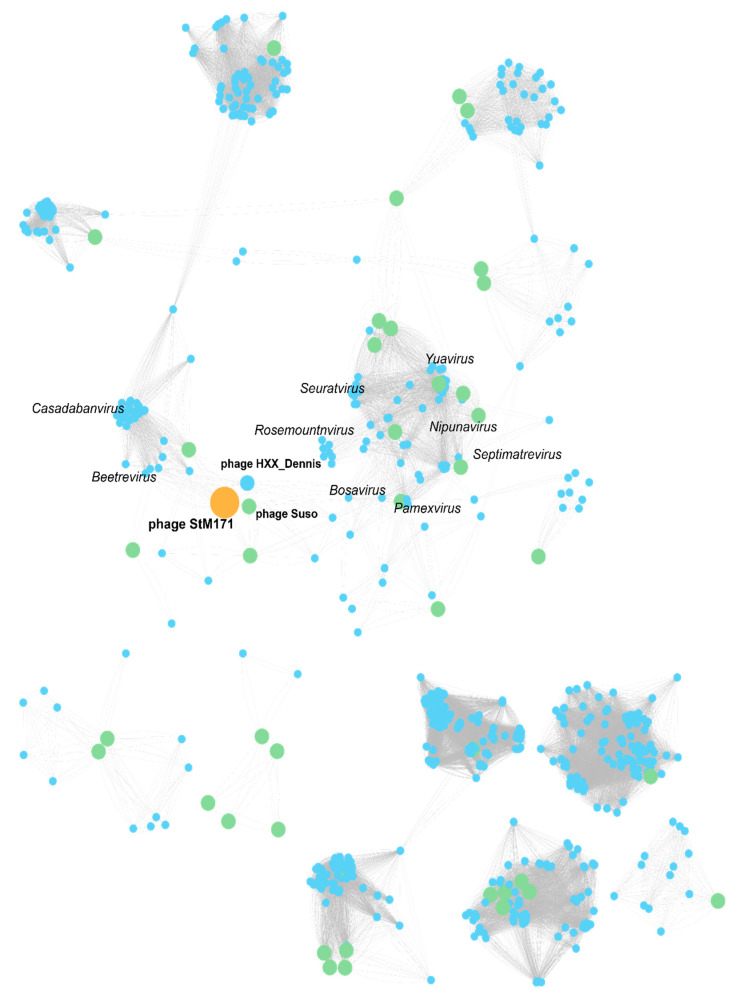
Protein-sharing network indicating the position of StM171 in relation to other phages. The network representation was produced with Cytoscape software (version 3.9.0) based on VConTACT2 calculations. The nodes indicate phage genomes, and edges between each two nodes indicate their statistically weighted pairwise similarities. The edge length is proportional to the similarity values estimated with the hypergeometric equation. Nodes are colored as follows: StM171 phage, orange; *Stenotrophomonas* phages, green; other phages, blue.

**Figure 7 viruses-15-02455-f007:**
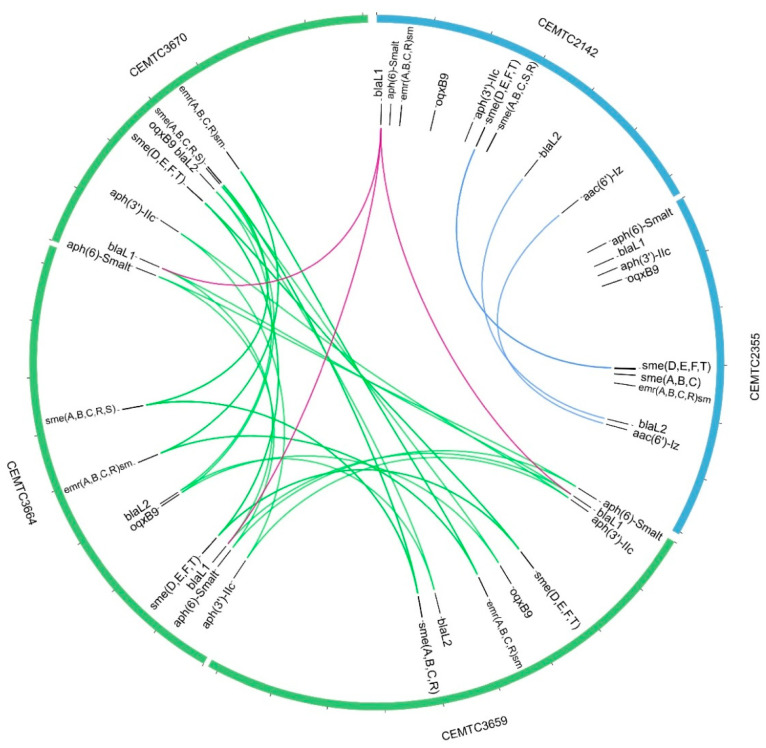
Diagram showing the genes responsible for antibiotic resistance found in StM171 bacterial host genomes with coverage of more than 80% in comparison with reference genes from different databases. The links between the genes represent a 100% identity. Blue links indicate genes identical among group A strains, green links indicate genes identical among group B strains, red links indicate genes identical among strains from both of group B and group A. The diagram was produced based on calculation performed using Abricate and AMRFinderplus modules in Bactopia workflow and visualized using Circos tool on Galaxy platform.

**Figure 8 viruses-15-02455-f008:**
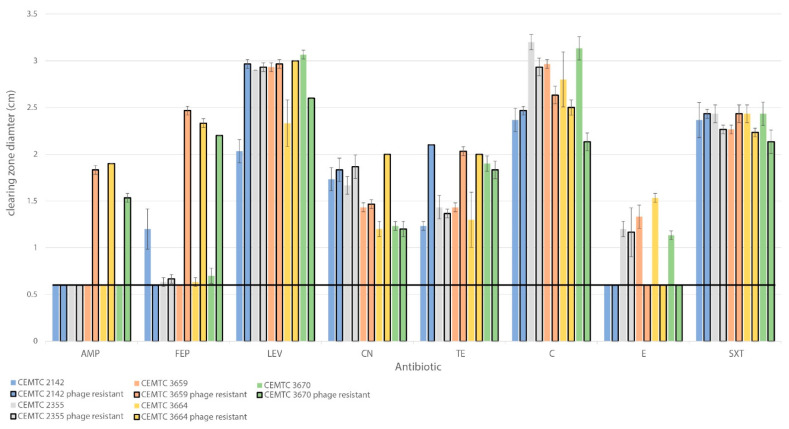
Applying antibiotics in disk diffusion assay to bacterial strains before and after they developed resistance toward StM171 phage. Each bolded column represents the phage-resistant bacterial strain. The black horizontal line represents the cut-off value based on the diameter in disk diffusion assay above which a strain is considered susceptible toward an antibiotic. Abbreviations: AMP: ampicillin, FEP: cefepime, LEV: levofloxacin, CN: gentamicin, TE: tetracycline, C: chloramphenicol, E: erythromycin, SXT: trimethoprim/sulfamethoxazole.

**Figure 9 viruses-15-02455-f009:**
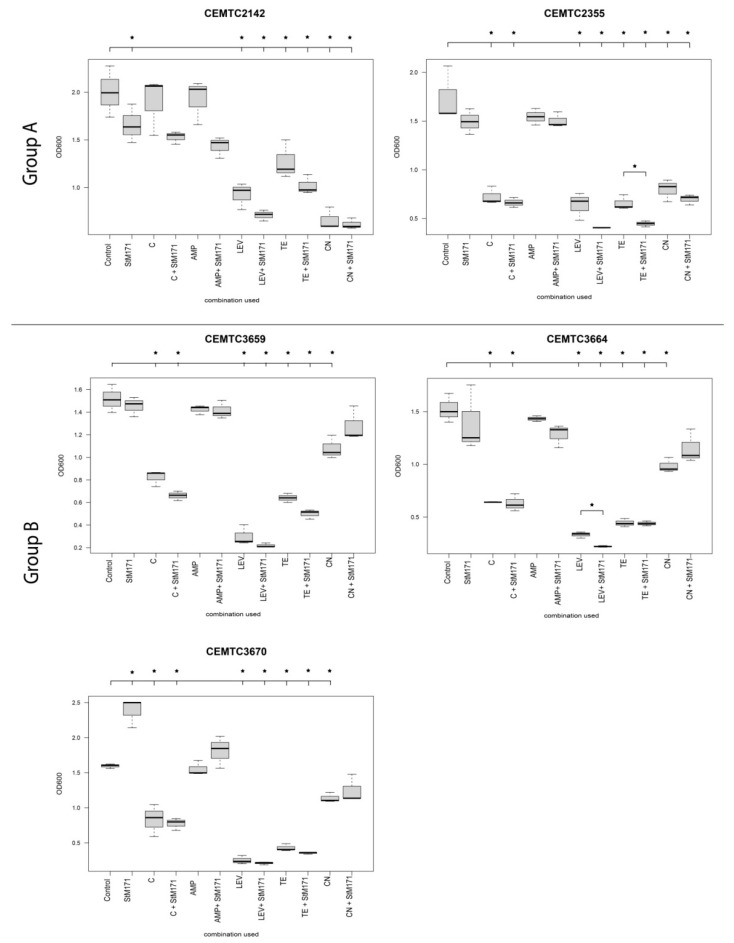
The preventive effect of using StM171 phage (MOI 0.1) in addition to various antibiotics (40% of MIC) to prevent the formation of biofilms by five different *S. maltophilia* strains. Abbreviations: Control: intact cell culture, C: chloramphenicol, AMP: ampicillin, LEV: levofloxacin, TE: tetracycline, CN: gentamicin, black stars indicate statistically significant changes in biofilm formation (*p* value < 0.05).

## Data Availability

Genome sequence of the *Stenotrophomonas* phage StM171 and 16S rRNA sequences of *Stenotrophomonas* strains are available in the GenBank database under the accession numbers: MZ611865, MZ424754, OP393915, MT040043, MT040044, and MT040045 for the phage StM171 and the strains *S. maltophilia* CEMTC 2142, CEMTC 2355, CEMTC 3659, CEMTC 3664, and CEMTC 3670, respectively.

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
