# Peer review of "StM171, a Stenotrophomonas maltophilia Bacteriophage That Affects Sensitivity to Antibiotics in Host Bacteria and Their Biofilm Formation"

_viruses, 2023, doi:10.3390/v15122455_

Round 1
Reviewer 1 Report
Comments and Suggestions for Authors
The manuscript describes the characterization of Stenotrophomonas maltophilia bacteriophage StM171 which belongs to a novel genus and affects antibiotics susceptibility and biofilm formation in the host bacteria.
Authors performed phage isolation and purification, analysis of phage genome and host range, bacterial host genome analysis focusing on the genotype related to antibiotic resistance and biofilm formation, assays for antibiotic resistance and for anti-biofilm activity.
The data presented in the paper are scientifically sound and clear, with no over-interpretations or problems, only a few minor edits to consider:
Line 79: please explain “ICBFM SB RAS” as it has been done for “CEMTC”
Line 102: “4â—¦C” please revise the use of “°” and make it uniform through the text
Line 122: please add a reference for software “image J”
Line 129: “min”, whereas line 136 it is “minutes”, please make it uniform through the text
Line 159: “de novo” requires italics
Line 160: “average coverage of 150” it is better to report “150x” or “150-fold”
Lines 162-171: please indicate the version of the tools used in the same way as in previous paragraphs (e.g. BLAST+ v2.12.0)
Line 175: “phages from NCBI GenBank database”, please provide info such as the number of phages added
Line 176: BLASTX search protein databases using a translated nucleotide query, so “to find the closest protein” rather than the closest genes? Please revise
Line 177: “Capsid” requires lowercase letter
Line 182: please provide the extensive name for “ICTV” and for “PCs”
Lines 191-194: these statements would fit better in the legend of figure 4
Lines 224 and 225: “disk” and “disc”, please uniform through the text
Line 278 and 281: please specify “Figure 2A” or “Figure 2B” in the text
Line 296: the word “ORFs” was already introduced at line 162, so the explanation can be omitted here
Line 298: “were considered hypothetical”, these ORFs likely encode hypothetical protein, therefore “encoded hypothetical proteins that showed no homology to other characterized sequences”. “G/C content” should be “GC content” like it is reported at line 170.
Line 299: “67,32” > “67.32”
Line 335: “a part of a clade” remove the first “a”
Line 381: proteins names should begin with uppercase letter, same for other protein names reported in the text
Line 382: “and CEMTC 2355”, please specify the subject (e.g. “and those of CEMTC 2355”)
Lines 384-387: these sentences are not fully clear. What is intended with the statement “the same set of the antibiotic resistance genes was found in the studied strains”? Also, line 384: “of the antibiotic resistance genes” removes “the”. Line 386-387: please specify nucleotide sequences (e.g. “showed variations in the nucleotide sequences”)
Line 392: “Abriacte” typo, please revise
Line 394: “A number of the antibiotic resistance genes were found”, please revise: either specify the number of antibiotic resistance gens found or just substitute “a number” with another expression
Line 369: Was an ICTV taxonomic proposal already made for this new genus?
Line 415-416: “lack of integration into host genomes” how did you assess the lack of prophage forms? Or this sentence was referring only to the presence of phage into the bacterial cell, if so please revise
Line 418: “compare” please adjust this verb for clarity, (e.g. “to compare with”)
Line 421: a “)” is missing before the “.”
Line 440: for clarity, “susceptible” specify susceptible to what. Also, specify that the strains used are the same five (e.g. “of the five susceptible”).
Line 445: “CEMTC2355” whereas at line 80 it is reported “CEMTC 2355”, please revise through the text and make it uniform
Line 465: “anti-biotics” please correct
Line 466: “S. maltophila” requires italics
Lines 488-491: a reference would be useful for this sentence
Table S1: please specify in a table note the reason why not all strains have an accession number
Table S4: for each primer, please add the nucleotide position on the phage genome. “STM” revise how it is written
Figure 2A: the letter “A” is present twice in the figure, please correct
Figure 2B: “one step growth curve of StM171 host strain S. maltophilia CEMTC 2355”, please change into “one step growth curve of StM171 on host strain S. maltophilia CEMTC 2355” for clarity
Author Response
We would like to thank the Reviewer for the detailed revision. We made all corrections according the comments.
Comments and Suggestions for Authors
The manuscript describes the characterization of Stenotrophomonas maltophilia bacteriophage StM171 which belongs to a novel genus and affects antibiotics susceptibility and biofilm formation in the host bacteria.
Authors performed phage isolation and purification, analysis of phage genome and host range, bacterial host genome analysis focusing on the genotype related to antibiotic resistance and biofilm formation, assays for antibiotic resistance and for anti-biofilm activity.
The data presented in the paper are scientifically sound and clear, with no over-interpretations or problems, only a few minor edits to consider:
Line 79: please explain “ICBFM SB RAS” as it has been done for “CEMTC”
- Explained
Line 102: “4â—¦C” please revise the use of “°” and make it uniform through the text
- Revised through the text
Line 122: please add a reference for software “image J”
- Added
Line 129: “min”, whereas line 136 it is “minutes”, please make it uniform through the text
- Corrected
Line 159: “de novo” requires italics
- Corrected
Line 160: “average coverage of 150” it is better to report “150x” or “150-fold”
Lines 162-171: please indicate the version of the tools used in the same way as in previous paragraphs (e.g. BLAST+ v2.12.0)
- We added the versions
Line 175: “phages from NCBI GenBank database”, please provide info such as the number of phages added
- 50 phages were added. We indicated the number in the paragraphs
Line 176: BLASTX search protein databases using a translated nucleotide query, so “to find the closest protein” rather than the closest genes? Please revise
- Revised
Line 177: “Capsid” requires lowercase letter
- Corrected
Line 182: please provide the extensive name for “ICTV” and for “PCs”
- We decrypted the abbreviations
Lines 191-194: these statements would fit better in the legend of figure 4
- These statements are related to figure 6, similar descriptive statements are mentioned in the legend of the figure 6
Lines 224 and 225: “disk” and “disc”, please uniform through the text
- Corrected
Line 278 and 281: please specify “Figure 2A” or “Figure 2B” in the text
- Specified
Line 296: the word “ORFs” was already introduced at line 162, so the explanation can be omitted here
Line 298: “were considered hypothetical”, these ORFs likely encode hypothetical protein, therefore “encoded hypothetical proteins that showed no homology to other characterized sequences”.
- Corrected
“G/C content” should be “GC content” like it is reported at line 170.
- Corrected
Line 299: “67,32” > “67.32”
- Corrected
Line 335: “a part of a clade” remove the first “a”
- Removed
Line 381: proteins names should begin with uppercase letter, same for other protein names reported in the text
- Corrected
Line 382: “and CEMTC 2355”, please specify the subject (e.g. “and those of CEMTC 2355”)
- Added
Lines 384-387: these sentences are not fully clear. What is intended with the statement “the same set of the antibiotic resistance genes was found in the studied strains”? Also, line 384: “of the antibiotic resistance genes” removes “the”. Line 386-387: please specify nucleotide sequences (e.g. “showed variations in the nucleotide sequences”)
- To infer the presence of the same genes of antibiotic resistance in all 5 studied strains, the sentence was slightly adjusted.
- “Seven antibiotic resistance genes were found in the five studied strains, the exception was the β-lactams resistance gene aac(6’)-Iz [66] that was found only in the group A strains”
Line 392: “Abriacte” typo, please revise
- Corrected
Line 394: “A number of the antibiotic resistance genes were found”, please revise: either specify the number of antibiotic resistance gens found or just substitute “a number” with another expression
- Indeed, there were 18 of them. We added the number
Line 369: Was an ICTV taxonomic proposal already made for this new genus?
- We are currently in the process of preparing the proposal
Line 415-416: “lack of integration into host genomes” how did you assess the lack of prophage forms? Or this sentence was referring only to the presence of phage into the bacterial cell, if so please revise
- We assessed the lack of prophage forms by conducting PCR screening for bacterial cells. To check for the presence of the StM171 genome in S. maltophilia bacterial strains, two sets of primers that targeted the genes encoding the capsid and tail assembly protein were designed. Primers are in (Table S4). Bacterial strains were heated at 95 °C for 10 minutes before centrifugation at 14000 rpm for 5 minutes and taking samples from the supernatant to be used as a template in PCR. It was described in the p. 2.13.
Line 418: “compare” please adjust this verb for clarity, (e.g. “to compare with”)
- Corrected
Line 421: a “)” is missing before the “.”
- Corrected
Line 440: for clarity, “susceptible” specify susceptible to what. Also, specify that the strains used are the same five (e.g. “of the five susceptible”).
- Corrected
Line 445: “CEMTC2355” whereas at line 80 it is reported “CEMTC 2355”, please revise through the text and make it uniform
- Was done
Line 465: “anti-biotics” please correct
- Corrected
Line 466: “S. maltophila” requires italics
- Corrected
Lines 488-491: a reference would be useful for this sentence
- Two references were added
Table S1: please specify in a table note the reason why not all strains have an accession number
- Sequences of 16S rRNA sequences from all maltophila strains were deposited. As for P. aeruginosa strains, 16S rRNA fragments were sequenced for most of them; however, we deposited 16S rRNA sequences in the GenBank database if the strain was isolated from new geographical place, new hospital/department (new patient, new infection). Just, a huge amount of 16S rRNA sequences from P. aeruginosa are in GenBank (and all of them are very close).
- Since no StM171-sensitive strains were found among aeruginosa strains, we removed sequences without Genbank number from the table and from the text.
Table S4: for each primer, please add the nucleotide position on the phage genome. “STM” revise how it is written
Figure 2A: the letter “A” is present twice in the figure, please correct
- Corrected
Figure 2B: “one step growth curve of StM171 host strain S. maltophilia CEMTC 2355”, please change into “one step growth curve of StM171 on host strain S. maltophilia CEMTC 2355” for clarity
- Corrected

Reviewer 2 Report
Comments and Suggestions for Authors
In their manuscript Jdeed et al describe new bacteriophage of Stenotropomonas maltophilia. The work is well carried out and well described and should be published, provide a few very minor points are adressed.
1. The host range of the phage is very narrow and the proliferation is very inefficent. Since the authors stat with the idea that this phage could be included into page therapy they should add a few sentences in the discussion whther this would be realy feasible unde thes conditions.
2. The Figure 1 is very impressive but additional confirmatory information would be provided when the authors also show a picture of a few additonal phages.
3. A controvery was discovered betwen the trees based on genome sequencing and the protein sequences of particular proteins. The authors argue on the different algorithms used which sound rather strange. Are there alternative interpretations?
4. Why were the novel antibiotics strains not sequenced?
Author Response
Reviewer 2:
We are grateful to the Reviewer for the positive review and the questions rose. To answer some of them, next investigation (project) should be started. We respond to every comment.
In their manuscript Jdeed et al describe new bacteriophage of Stenotropomonas maltophilia. The work is well carried out and well described and should be published, provide a few very minor points are adressed.
- The host range of the phage is very narrow and the proliferation is very inefficent. Since the authors stat with the idea that this phage could be included into page therapy they should add a few sentences in the discussion whther this would be realy feasible unde thes conditions.
The phage studied in this paper is not intended for phage therapy use, it was studied as a model to show the ability of even weak lytic phage to restore sensitivity of bacterial hosts toward antibiotics. We point out the inapplicability of this phage for phage therapy in the last paragraph.
- The Figure 1 is very impressive but additional confirmatory information would be provided when the authors also show a picture of a few additonal phages.
We can add another picture to the supplementary materials
- A controvery was discovered betwen the trees based on genome sequencing and the protein sequences of particular proteins. The authors argue on the different algorithms used which sound rather strange. Are there alternative interpretations?
In addition to the difference of the algorithm used, an explanation is that phages from the proposed new genus are characterized by mosaicism. In addition, the evolution of this genus is not clear and the evolution should be investigated in special detailed study.
- Why were the novel antibiotics strains not sequenced?
We plan to sequence them in an upcoming project.
